# LIG-Based High-Sensitivity Multiplexed Sensing System for Simultaneous Monitoring of Metabolites and Electrolytes

**DOI:** 10.3390/s24216945

**Published:** 2024-10-29

**Authors:** Sang Hyun Park, James Jungho Pak

**Affiliations:** School of Electrical Engineering, Korea University, Seoul 02841, Republic of Korea; yh99gosh22@korea.ac.kr

**Keywords:** wearable biosensor, laser-induced graphene, glucose sensor, lactate sensor, Na^+^ sensor, K^+^ sensor

## Abstract

With improvements in medical environments and the widespread use of smartphones, interest in wearable biosensors for continuous body monitoring is growing. We developed a wearable multiplexed bio-sensing system that non-invasively monitors body fluids and integrates with a smartphone application. The system includes sensors, readout circuits, and a microcontroller unit (MCU) for signal processing and wireless communication. Potentiometric and amperometric measurement methods were used, with calibration capabilities added to ensure accurate readings of analyte concentrations and temperature. Laser-induced graphene (LIG)-based sensors for glucose, lactate, Na^+^, K^+^, and temperature were developed for fast, cost-effective production. The LIG electrode’s 3D porous structure provided an active surface area 16 times larger than its apparent area, resulting in enhanced sensor performance. The glucose and lactate sensors exhibited high sensitivity (168.15 and 872.08 μAmM^−1^cm^−2^, respectively) and low detection limits (0.191 and 0.167 μM, respectively). The Na^+^ and K^+^ sensors demonstrated sensitivities of 65.26 and 62.19 mVdec^−1^, respectively, in a concentration range of 0.01–100 mM. Temperature sensors showed an average rate of resistance change per °C of 0.25%/°C, within a temperature range of 20–40 °C, providing accurate body temperature monitoring.

## 1. Introduction

As the medical environment has advanced and interest in improving the quality of life has increased within the general public, treatment methods have moved on from a passive method to an active method by using devices (such as wearable biosensors) that continuously monitor our bodily conditions [1,2,3]. These wearable biosensors can continuously monitor various health-related parameters without restricting the subject’s movement by direct attachment to the subject’s body [4,5,6,7,8,9,10,11,12,13]. In addition, the increase in smartphone usage has also been closely tied to the development of wearable devices, owing to the ease and accessibility these phones provide to their users [14,15,16,17,18,19,20,21,22]. As most of the initial wearable sensors mainly consisted of physical sensors designed to monitor heart rate and body movement, there was a clear need for wearable sensors that could monitor our bodily health on a more comprehensive level [4,14,20]. Currently, research is ongoing to develop biosensors and wearable devices that can monitor and gleam medically relevant information from numerous biomarkers that exist in biofluids such as blood, interstitial fluid (ISF), sweat, and tears using electrochemical methods [14,20,23,24,25,26]. The commercialization of wearable biosensors using these electrochemical methods is underway and has grown significantly in the last five years. According to Grand View Research Inc. (San Francisco, CA, USA), it is a promising research field with an expected compound annual growth rate (CAGR) increase of 38.8% and will reach a market value of USD 27.8 billion by the year 2025 [27,28].

Currently, electrochemical wearable biosensors are evolving toward monitoring various physiological information through biofluids that can be collected non-invasively. Previously, most biosensors using electrochemical measurement methods have targeted blood. However, these methods of using blood are invasive, leading to multiple problems such as user pain, risk of infection, and low sampling rate. Therefore, it is crucial that wearable biosensors aimed at real-time monitoring use non-invasive methods of accessing biological fluids. Among the various biofluids available for non-invasive methods, interstitial fluid (ISF) has biomarker concentrations similar to that of blood, making it possible to do previous blood-related assays on ISFs instead. ISF can be collected without damaging the skin by electrophoresis but has the disadvantage of having difficulty controlling the amount of discharge in the case of heavy molecules such as glucose [14]. Tears can be measured using a contact lens-type wearable sensor, but the design or measurement method has strict limitations because tears cannot be artificially induced and require direct contact with the eye [27,29,30,31,32,33,34,35]. Diabetic patients’ saliva glucose concentration has a high correlation with various biomarkers, such as alcohol and salinity, that can be measured, but sensor contamination may occur due to food debris or high concentrations of protein in saliva [36,37,38,39,40,41,42]. Sweat can be easily acquired from any part of the body due to the widespread distribution of sweat glands throughout the entire body, making it advantageous to be used as a measurement target of wearable sensors [20,43].

Although not in the form of a wearable sensor, a method of monitoring health status using sweat has been developed and tested before. In the case of cystic fibrosis, which damages the lung, the chloride-reuptake channel of the sweat gland malfunctions, making it possible to diagnose cystic fibrosis by measuring the concentration of Cl^−^ in sweat [20,44]. In addition, biomarkers such as various electrolytes (Na, K, Ca, NH_4_) and metabolites (glucose, lactate, urea, ethanol, cortisol) in sweat can be used as an index to check various physiological information. For example, in sports, sweat can be used for doping tests by measuring drug metabolites in sweat [20,45,46,47]. Despite these advantages, conventional sweat collection and analysis require expensive and time-consuming procedures. In addition, during sweat collection, reliability may be an issue due to evaporation or chemical decomposition by the time measurement is underway.

Here, we report a highly sensitive multiplexed sensing system that can detect glucose, lactate, sodium ions, potassium ions, and temperature simultaneously. Previous studies using 2D materials had the drawback of requiring a vacuum environment for the lithography process during electrode patterning, which made the electrode fabrication process time-consuming [48,49]. However, using laser-induced graphene (LIG) allows for rapid electrode patterning, and its 3D porous structure provides a larger active surface area that directly contributes to sensing, enhancing sensitivity. For these reasons, LIG was chosen for sensor fabrication. Sensors that can selectively measure glucose, lactate, sodium ions, and potassium ions using enzymes or ion-selective membranes (ISM) were fabricated. It was confirmed that the glucose and lactate sensors had very high sensitivity and low limit of detection (LOD) due to the 3D porous structure of the LIG electrode. We have produced a small, mobile printed circuit board (PCB) that can process electrical signals from multiple sensors simultaneously and wirelessly transmit the processed data to a smartphone. The PCB includes a readout circuit designed specifically for each sensor, a microcontroller unit (MCU) for signal processing, and a wireless communication device. The operation of the board can be controlled through a smartphone connected to the board, and a calibration mode has been added for more accurate measurements. During measurement, the output value of the sensors can be visualized in real time, and the measured values can be converted to the concentration and temperature of each substance. All measured data can be saved, and after the measurement is completed, the signal or concentration and temperature change according to time can be plotted. The highly sensitive multiplexed sensing system can act as a solution for the continuous monitoring of biomarkers present in bodily biofluids.

## 2. Materials and Methods

### 2.1. Fabrication of LIG-Based Multisensor Array

#### 2.1.1. Design and Fabrication of a Multisensor Array and Passivation Layer

The multisensor array and passivation layers are shown in Figure 1a. The LIG-based temperature sensor was placed in the upper center of the multisensor array with a line width of 0.25 mm and a total length of 36 mm in right-angle serpentine form. The diameter of the fabricated circular working electrodes (WE) of all the glucose, lactate, Na^+^, and K^+^ sensors was 1.2 mm, and the diameter of its exposed area in the protective passivation layer was 1.0 mm. The rectangular reference electrodes (RE) of the glucose, lactate, Na^+^, and K^+^ sensors each had a width of 0.4 mm and a height of 1.5 mm, and the open area of the RE on the protective layer had a 0.3 mm width and a 1.4 mm height.

Figure 1b shows the LIG electrode formation process, and Figure 1c shows the finished LIG electrode on the PI film. LIG electrodes were formed by irradiating a 250 μm-thick PI film (DuPont Inc., Wilmington, DE, USA) with a CO_2_ laser (VLS2.30, Universal Laser System Inc., Scottsdale, AZ, USA) using raster mode at 1000 PPI (pulse per inch), 1000 DPI (dot per inch), a power of 4.2 W, wavelength of 10.6 µm, and scan speed of 88.9 mm/s. After LIG electrode formation on the PI film, carbon particles on top of the electrodes left over from the fabrication process were removed by blowing with N_2_ gas. The passivation layer was fabricated by CO_2_ laser irradiation in vector mode on a 55 μm thick PI tape (DuPont Inc., Wilmington, DE, USA) to open the WEs and contact pads of the glucose, lactate, Na^+^, and K^+^ sensors. Figure 1d shows the structure of the multisweat sensor.

#### 2.1.2. Fabrication of Working Electrodes of Multi Sweat Sensor

Each of the glucose, lactate, Na^+^, and K^+^ sensors had its own pair of WEs and REs. Among them, the WEs of the glucose and lactate sensors were fabricated by electrodepositing PdCu and immobilizing GOx and LOx, respectively, on the exposed area of LIG WEs. GOx enzyme reacts with glucose to generate hydrogen peroxide, which is then decomposed effectively by a commonly used PdCu catalyst, resulting in the enhanced output signal with higher sensitivity. On the other hand, the LOx enzyme reacts with lactate to generate pyruvate, which is not catalyzed by PdCu. However, the output current may be enhanced because of the lower resistance of the LIG WE when electrodeposited. PdCu compared to that without PdCu electrodeposition. PdCu was electrodeposited by cyclic voltammetry (CV) for five cycles in the range of −0.8~0.2 V using a commercial graphite electrode as CE and a commercial Ag/AgCl electrode as RE. PdCu electrodeposition solution was prepared by combining and stirring 0.1 M HClO_4_, 7 mM PdCl_2_, and 3 mM CuCl_2_ solution for 2 h [50]. The glucose WE fabrication is completed after drop-casting 1.5 μL of GOx immobilization solution and drying it. The GOx immobilization solution was prepared by stirring 1 mL of 50 mM acetic acid and 1 wt% chitosan at room temperature for 1 h and then adding 50 mg of GOx, 17 mg of bovine serum albumin (BSA), and 17 mg of ectoine. The GOx immobilization solution was then drop-casted on the PdCu-electroplated electrode and immobilized at 4 °C for 12 h to complete the glucose WE. The lactate WE can also be fabricated using the aforementioned same process as the glucose sensor fabrication, except that LOx was used in place of GOx, and 17 mg of ectoine was not used [4].

Na^+^ and K^+^ WEs were fabricated by poly(3,4-ethylenedioxythiophene) polystyrene sulfonate (PEDOT: PSS) electropolymerization as an ion-electron converter to minimize potential drift and then an ion-selective membrane (ISM) was layered on LIG electrodes. PEDOT: PSS was electropolymerized by submerging the LIG WE in 10 mM 3,4-ethylenedioxythiophene (EDOT) and 0.1 M Na polystyrene sulfonate (NaPSS) while stirring and applying 2 mAcm^−2^ (vs. commercial Ag/AgCl) for 715 s. The Na ISM cocktail was prepared by adding 660 μL of tetrahydrofuran to 100 mg of a mixture of Na ionophore X 1 wt%, Na-TFPB (tetrakis[3,5bis(trifluoromethyl)phenyl]borate) 0.55 wt%, PVC(polyvinyl chloride) 33 wt%, and DOS (bis(2-ethylehexyl) bebacate) 65.45 wt%. This was drop-casted onto the PEDOT: PSS electropolymerized electrode and dried at room temperature for 24 h to produce the Na^+^ sensitive WE. K ISM cocktail was prepared by adding 350 μL of cyclohexanone to 100 mg of a mixture of 2 wt% valinomycin, 0.55 wt% Na-TFPB, 32.7 wt% PVC, and 64.7 wt% DOS. This was drop-casted onto the PEDOT: PSS electropolymerized electrode and dried for 24 h at room temperature to produce the K^+^ sensitive WE. REs of glucose, lactate, Na^+^, and K^+^ sensors were all coated with Ag/AgCl ink (BAS Inc., Tokyo, Japan) on the LIG electrode and cured for 20 min at 80 °C in a dry oven. Modified solid-state Ag/AgCl REs were fabricated by drop-casting 4 μL of a mixture of 79.1 mg PVB, 50 mg NaCl, and 1 mL methanol. Modified solid Ag/AgCl REs were not significantly affected by different Cl^−^ concentrations in the test samples. As shown in Figure 1b, the temperature sensor was fabricated in a zigzag form of a LIG electrode 0.25 mm wide and 36 mm long.

### 2.2. Design of Multiplexing System and Smartphone Application

#### 2.2.1. Design of Multiplexing System: Readout Circuits

Figure 1e shows the completed multisensor and multiplexing system, and Figure 1f shows the diagram of the multiplexing system. This multiplexing system consists of three major components. The first component is the readout circuit that converts and conditions the output signal of each sensor to be recognized by a microcontroller unit. The glucose and lactate sensors share the same amperometry readout circuit. The Na^+^ and K^+^ sensors share the same open-circuit potential readout circuit. Since four electrochemical sensors are supposed to operate simultaneously in one solution, the applied voltage could affect the measurement output of other sensors and generate inter-sensor crosstalk. To mitigate this crosstalk, the amperometry measurements were performed at an applied voltage of 0 V. While the measured output signal from the fabricated glucose and lactate sensors is in the form of negative current, the selected MCU can accept only positive voltage values. Therefore, the negative current signal was converted into negative voltage using a trans-impedance amplifier and then inverted by an inverting amplifier to produce positive voltage so that the signal could be recognized by the MCU. The feedback resistance value of the trans-impedance stage was selected considering the dynamic output range of each sensor within the concentration range of glucose and lactate in human sweat and the voltage range that could be processed by the op-amps and MCU. The Na^+^ and K^+^ sensor output signals were measured by potentiometry, and the potential difference between WE and RE was determined by a differential amplifier. Also, considering the input voltage ranges of the MCU (0–5.0 V) and Op-Amp (0–3.5 V), the gain was set to 4, and the DC offset was set to 1.5 V for the trans-impedance stage. As for the temperature sensor, the resistance change of the LIG electrode due to temperature variation was measured using a voltage divider circuit. The temperature sensor (LIG electrode) and 1 kΩ resistor were connected in series, with the open end of the resistor being connected to the ground and the open end of the temperature sensor being connected to a 5 V power supply. The temperature was calculated by calculating the measured resistance value of the temperature sensor.

#### 2.2.2. Design of Multiplexing System: MCU and Bluetooth Module

The second component is an MCU that simultaneously receives analog signals processed by the readout circuits of each sensor, converts them into digital signals, and transmits the data to the Bluetooth module so they can be sent to a smartphone. ATmega328p MCU, which is commonly used by low-power and low-cost embedded systems, was used to convert the analog voltage signals processed by the readout circuits into digital signals through the 10-bit analog-to-digital converter (ADC) built into the MCU. In addition, the contents converted from the readout circuit were inverted and converted into the output signal of each sensor. The output signal is then sent by a Bluetooth wireless transmission module to a smartphone.

The third component is the Bluetooth module. We used the FBL780BC BLE with an average current consumption of 2 μA in the connected state to configure a low-power system. In addition, a +3.7 V lithium-ion battery (power source), voltage regulators (to convert +3.7 V to +5 V, −5 V, 3.3 V), an LM324 IC chip (quad OP-AMP), and a five-pin type USB charger (for charging the +3.7 V lithium-ion battery) were used. As shown in Figure 1e, the multiplexing system is composed of two (top and bottom) PCBs, which were interconnected by 2.54 mm pin headers and sockets. The bottom PCB contains the ZIF connector and readout circuit. The top PCB consists of the MCU, Bluetooth module, regulators, and battery charger module.

#### 2.2.3. Design of Smartphone Application for Real-Time Data Manipulation and Display

The smartphone application was created to receive, display, and save data in real time within an Android device. The application was created using the MIT app inventor for the Android operating system. The application has two modes: one displays the output value of each sensor, and the other displays the corresponding concentrations and temperature of the measured substance. Once the application is connected to the multiplexing system via Bluetooth, the ‘measure’ button is pressed to continue the measurement. During measurement, each sensor’s output data appear as shown in Appendix A. If the ‘stop’ button is pressed during the measurement, the measurement will end, and the measured data are saved in CSV format in the file location shown at the bottom of Appendix A. To plot the data from each sensor into a graphical form, the ‘plot’ button is pressed, as shown in Appendix A. As shown in Appendix A, the calibration is performed by touching the ‘calibration 1’ button while the sensor is in the minimum concentration solution and then touching the ‘calibration 2’ button while the sensor is in the maximum analyte concentration solution. The ‘measurement mode’ button on the upper right side is touched to calculate the sensitivity of the sensor and to display the concentration of the measured target analyte.

### 2.3. Characterizations and Performance Measurement Methods of Sensors and Multiplexed Sensing System

#### 2.3.1. Active Surface Area of LIG Electrode

To examine the active surface area of the LIG electrode, the peak current was measured by cyclic voltammetry (CV) depending on the scan rate. CV was conducted ion-electron converter in 5 mM K_3_[Fe(CN)_6_] solution at a scan rate of 10, 20, 30, 50, 100, and 200 mVs^−1^ using a commercial potentiostat (Gamry (Warminster, PA, USA), PCI4/750 ™). The active surface area was calculated using the Randles-Sevcik equation with the peak current value according to the scan speed obtained by CV.

#### 2.3.2. Characterizations of PVB-Coated LIG-Based Ag/AgCl RE

The potential difference between PVB-coated LIG-based Ag/AgCl RE and a commercial Ag/AgCl RE was measured in varying NaCl concentrations to examine whether the Ag/AgCl electrode maintained a constant potential regardless of the Cl^−^ ion concentrations in the test samples. The measurement was performed using a commercial potentiostat in an aqueous solution of NaCl at concentrations ranging from 10^−1^ to 10^−4^ M in potentiometric mode.

#### 2.3.3. Sensitivity of the Proposed Glucose Sensor and Lactate Sensor

The proposed glucose sensors and lactate sensors have been successfully fabricated, and their performance has been analyzed by measuring the outputs in the physiologically relevant concentration ranges of glucose and lactate in human sweat. The measurement was carried out using a commercial potentiostat at the reading voltage set to 0 V using chrono-amperometry with the PVB = coated Ag/AgCl RE and CE. It was reported that the glucose and lactate concentrations present in human sweat typically range 0.2–0.6 mM and 9–23 mM, respectively [51,52]. Therefore, test solutions for the glucose and lactate sensors were prepared at ranges of 0–3 mM and 0–30 mM, which are slightly broader than the typical ranges in human sweat for glucose and lactate, respectively. Glucose solutions at concentrations of 0, 0.01, 0.1, 0.2, 0.3, 0.5, 1.0, 2.0, and 3.0 mM and lactate solutions at concentrations of 0, 0.1, 0.2, 0.3, 0.5, 1.0, 2.0, 3.0, 5.0, 10.0, 20.0, and 30.0 mM were prepared.

#### 2.3.4. Sensitivity of the Proposed Na^+^ Sensor and K^+^ Sensor

The proposed ion sensors have been successfully fabricated, and their outputs were measured at various concentrations of Na^+^ and K^+^ in order to analyze the sensor performance. The fabricated Na^+^ and K^+^ sensors were tested in Na^+^ and K^+^ concentrations that were reported to exist in human sweat. The open-circuit potential difference between WE and RE of the ion sensor was measured using a commercial potentiostat. The typical Na^+^ and K^+^ concentration ranges in human sweat are known to be 10–90 and 2–10 mM, respectively [53]. Therefore, NaCl and KCl solutions were prepared with concentrations at 0.01, 0.1, 1.0, 10.0, and 100.0 mM.

#### 2.3.5. Selectivity of Glucose, Lactate, Na^+^, and K^+^ Sensors

Various electrolytes and metabolites in sweat may act as interferences when measuring glucose, lactate, Na^+^, and K^+^. Appendix A shows the composition and concentration of the solutions of different substances prepared for each sensor to see their responses to various interference substances. To investigate the selectivity of glucose and lactate sensor, a mixture of acetic acid (AA), uric acid (UA), glucose, lactate, sucrose, NaCl, and KCl was added to 1× PBS. The applied voltage was set to 0 V, and the amperometric current was measured at 50 s for each substance. For the selectivity of the Na^+^ and K^+^ sensor, a mixture of NaCl, KCl, NH_4_Cl, MgCl_2_, CaCl_2_, and lactate was added to DI water. The potential difference between WE and RE was measured for 50 s in the solutions shown in Appendix A using a commercial potentiostat.

#### 2.3.6. Characterization of Temperature Sensor

Glucose, lactate, Na^+^, and K^+^ measurements are all influenced by temperature, thereby necessitating the existence of the temperature sensor for calibration. The resistance change of the temperature sensor was measured using a commercial source meter (Keithley 2400) while controlling the environment temperature using a hot plate. To confirm the performance of the temperature sensor, artificial sweats of different temperatures were prepared. The resistance of the temperature sensor was measured while increasing the temperature setting of the hot plate by 5 °C from 20 to 40 °C. The actual temperature of the hot plate surface measured by a commercial temperature sensor (TOYOTECH) was measured at 23.3, 27.6, 30.2, 35.0, 39.6, and 43.7 °C.

#### 2.3.7. Characterization of Multisensor Array and Multiplexed Sensing System in Artificial Sweat

Artificial sweat was prepared with a mixture of glucose, lactate, Na^+^, and K^+^ in addition to various electrolytes (NH^4+^, Ca^2+^, and Mg^2+^) and metabolites (Urea, Uric acid) that are found in natural sweat. Artificial sweat was prepared in five different concentrations of the target analytes shown in Appendix A. To verify the operation of the final LIG-based multi-electrochemical sensor system, the LIG-based electrodes were combined with the readout PCB and the wireless communications system along with the smartphone application. To verify the operation of the LIG-based multi-electrochemical sensor, integrated PCB, and smartphone application, measurements were conducted at different concentrations and temperatures of artificial sweat. First, to measure the concentration of analytes in artificial sweat, calibration 1 was performed on the lowest concentration of artificial sweat, and calibration 2 was performed on the highest concentration of artificial sweat, shown in Appendix A. After calibration, target analyte concentrations in artificial sweat, excluding the lowest and highest analyte concentrations, were measured. To confirm the operation of the temperature sensor, the artificial sweat of the same concentration was heated to 25, 30, 35, 40, and 45 °C for 30 min on a hot plate, and then the temperature was measured. The temperature of artificial sweat measured by a commercial temperature sensor was 24.5, 30.3, 35.4, 40.2, and 45.1 °C. The temperature sensor was calibrated by performing calibration 1 at the lowest temperature and calibration 2 at the highest temperature. After calibration, the temperature was measured in artificial sweat at three different temperatures. The measurements were performed for 50 s in all solutions, and the measurements were paused when the solution was changed.

## 3. Results and Discussion

The goal of this research is to develop a sensor system that consists of four parts: (1) sensors that can measure temperature and the concentrations of glucose, lactate, Na^+^, and K^+^ in artificial sweat, (2) readout circuits, (3) wireless data communications module, and (4) a mobile application to control the sensor function, save the data, and display the results. We have investigated glucose, lactate, Na^+^, and K^+^ monitoring performance within the relevant ranges of a typical human sweat in addition to the influence of interference substances normally found in human sweat. In addition, the correlation between sensor output (amperometric current for glucose and lactate, potential for Na^+^ and K^+^) and target analyte concentration, in addition to resistance change in varying temperatures in artificial sweat, was investigated.

### 3.1. Characterizations of LIG Electrodes and PVB-Coated Ag/AgCl RE

#### 3.1.1. Characterizations of LIG Electrodes

To analyze the structure of the LIG electrode grown on the PI film, FE-SEM (FEI, Quanta 250FEG) and Raman (HORIBA (Kyoto, Japan), LabRam ARAMIS IR2) spectroscopy analyses were performed. Figure 2 shows FE-SEM photographs of the LIG electrode Figure 2a top-view and Figure 2b cross-section. Figure 3a shows the Raman spectroscopy results. A D peak at 1350 cm^−1^, a G peak (commonly found in graphite-based materials) at 1580 cm^−1^, and a 2D peak at 2700 cm^−1^ were observed. The D peak is caused by inelastic scattering by a phonon, and the higher the intensity of the peak, the more defects. The 2D peak appears at 2700 cm^−1^, which is twice that of 1350 cm^−1^ because inelastic scattering by phonons with 1350 cm^−1^ energy appears twice. The number of graphene layers can be deduced by obtaining I_2D_ (intensity of 2D peak)/I_G_ (intensity of G peak). An I_2D_/I_G_ value of 1.8 means a monolayer; when between 1.8 and 0.8, it means a few layers, and below 0.8, it means a multilayer structure [54]. In this study, the I_2D_/I_G_ was 0.83, meaning the LIG electrode had a structure of a few stacked layers of graphene. To confirm the active surface area of the LIG electrode, the peak current according to the scan rate was measured by cyclic voltammetry (CV) and calculated using the following Randles–Sevcik Equation (1) [55].
(1)$$I_{p}=\left(2.69\times 10^{5}\right)AD^{\frac{1}{2}}\;n^{\frac{3}{2}}\;Cv^{\frac{1}{2}}$$

Here, *I_p_* is the peak current, *A* is the active surface area, *D* is the diffusion constant of the molecule in the measurement solution ($6.70\;\times \;10^{-6}\;{cm}^{2}s^{-1}$ in 5 mM K_3_[Fe(CN)_6_] solution), *n* is the number of electrons participating in the redox reaction, *C* is the concentration of the molecule in the redox solution (mol cm^−3^), and *v* is scan rate (Vs^−1^). The peak current value was calculated by substituting scan speed into the Randles–Sevcik equation. As a result, the active surface area of the 1 mm diameter LIG electrode was 12.7 mm^2^, which was 16 times larger than the layout area of 0.785 mm^2^. Figure 3b shows the CV measurement result, and Figure 3c shows the peak current depending on the scan speed.
Figure 2FE-SEM images of LIG electrode. (**a**) Top view. (**b**) Cross-sectional view.
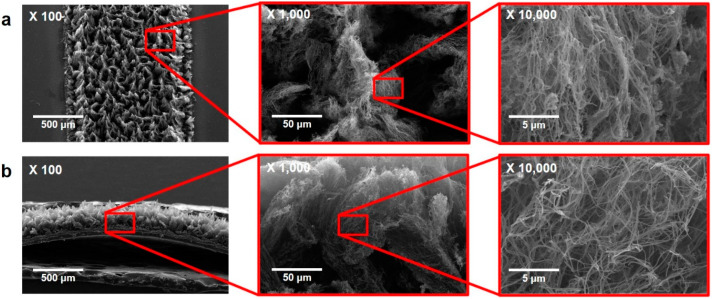

Figure 3(**a**) Raman spectrogram of PI film and LIG. (**b**) Raw CV plot and (**c**) peak current plot of LIG electrode according to scan speed. (**d**) Plot of potential difference with and without PVB coating Ag/AgCl RE vs. commercial Ag/AgCl at various NaCl concentrations.
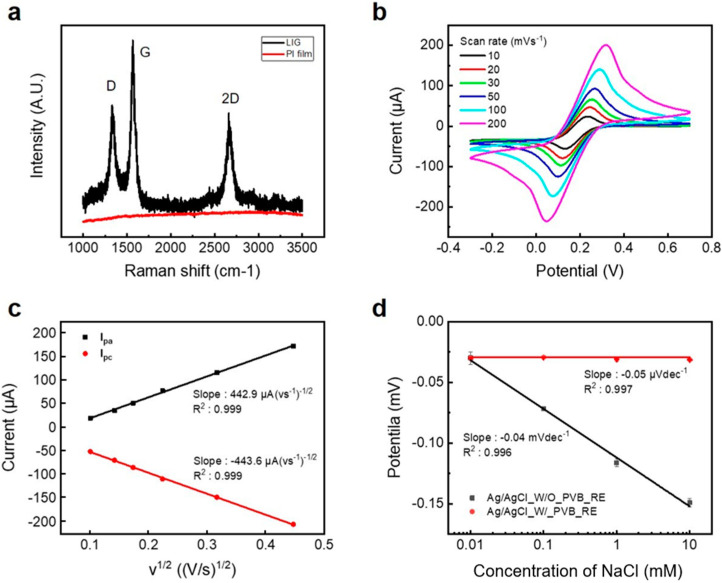


#### 3.1.2. Effect of PVB Coating on LIG-Based Ag/AgCl RE

The glucose, lactate, Na^+^, and K^+^ sensors share a common Ag/AgCl reference electrode. The potential difference between PVB-coated LIG-based Ag/AgCl RE and a commercial Ag/AgCl RE was measured in various NaCl concentrations to investigate whether the Ag/AgCl electrode maintains a constant potential regardless of the Cl^−^ ion concentrations in the test samples. The measurement was performed using a commercial potentiostat in an aqueous solution of NaCl at a concentration ranging from 10^−1^ to10^−4^ M in potentiometric mode. Figure 3d shows the potential between the Ag/AgCl electrode with or without PVB coating and the commercial Ag/AgCl electrode in varying NaCl concentrations. The Ag/AgCl electrode without PVB had a sensitivity of 0.04 mVdec^−1^ to NaCl, whereas the Ag/AgCl electrode coated with PVB had a much lower sensitivity of 0.05 μVdec^−1^, indicating greater stability even in the presence of vastly different concentrations of Cl^−^ ions.

### 3.2. Functional Analysis of LIG-Based Multisensor Array

#### 3.2.1. Sensitivity of LIG-Based Multisensor Array

Appendix A show the currents generated from the glucose and lactate sensors, respectively. Figure 4a,b show the calibration plots based on the data of Appendix A and show the amperometric currents vs. concentrations of glucose and lactate. The slopes of the current values according to the concentrations measured at 0–3.0 mM (R^2^: 0.999) and 0–30.0 mM (R^2^: 0.991), respectively, using glucose and lactate sensors were −1.32 μAmM^−1^ and −4.73 μAdec^−1^. The diameter of the WE open area is 1 mm, and the sensitivity of the glucose and lactate sensors per unit area were 168.15 μAmM^−1^cm^−2^ and 602.54 μAdec^−1^cm^−2^, respectively. Figure 4c and Appendix A show the sensitivity and linearity when the sensitivity unit is in μAmM^−1^ instead of μAdec^−1^ for comparison with the performance of lactate sensors in other studies. Here, the linearity was 0.966 from 0–0.5 mM, and the sensitivity per unit area was 872.08 μAmM^−1^cm^−2^. Figure 4d and Appendix A show the amperometric current response depending on the lactate concentration in the range of 0–0.5 mM. Figure 4b,d both present results for the lactate sensor, but they cover different measurement ranges, with Figure 4d showing results at lower concentrations.

The limit of detection (LOD) was calculated as LOD = 3σ/b, where σ is the standard deviation of the measured value at the lowest concentration, and b is the sensitivity of the measured analyte. The calculated LOD of the glucose sensor was 0.191 μM, and the LOD of the lactate sensor was calculated to be 0.167 μM.

Figure 5a,b compare the sensor characteristics of this study against other studies related to enzymatic electrochemical glucose and lactate sensors, respectively. Our LIG-based glucose sensor showed the lowest LOD and the fourth highest sensitivity among the reported sensors that are described in the 17 references. In addition, our LIG-based lactate sensor not only showed the lowest LOD among the 12 sensors but also showed the highest sensitivity. Figure 5c,d show the potential difference between the Na^+^ sensor’s WE and RE and the potential difference between the K^+^ sensor’s WE and RE. The Na^+^ and K^+^ sensors showed a sensitivity of 65.26 mVdec^−1^ and 62.19 mVdec^−1^ from the concentration range of 0.01 to 100 mM of Na^+^ and K^+^, respectively. Compared to the theoretical Nernstian value of 59.16 mV dec^−1^, the obtained sensitivities for Na^+^ and K^+^ sensors are 6.1 and 3.03 mV dec^−1^ higher.

#### 3.2.2. Selectivity of Multisensor Array

Figure 6 shows the amperometric current response of the glucose (Figure 6a) and lactate (Figure 6b) sensors and the open-circuit potential (OCP) of the Na^+^ (Figure 6c) and (d) K^+^ (Figure 6d) sensors. The glucose and lactate sensors showed a sensitivity of less than 2 nAmM^−1^ when a substance other than glucose or lactate was added. The Na^+^ and K^+^ sensors showed a very low sensitivity of 8 μVmM^−1^ or less when substances other than Na^+^ or K^+^ (such as NH_4_Cl, MgCl_2_, CaCl_4_, etc.) were added. Each sensor showed that the output signal change for the target analytes was eight times higher than that of interference substances. The concentrations of the added interfering substances are not identical because we aimed to maintain the concentration ratios found in sweat [56,57].

#### 3.2.3. Characterization of Temperature Sensors

Appendix A shows the resistance of the temperature sensor according to the temperature of the commercial temperature sensor’s measurement temperature. Appendix A shows the average and standard deviation measured across five sensors. The average resistance change rate was 5.10% in the range of 20–40 °C, and the average resistance change rate per °C was 0.25%/°C.

### 3.3. Characterization of Multiplexing System

#### 3.3.1. Analysis of Artificial Sweat Depending on Analyte Concentration and Temperature

Appendix A show that the measured outputs change among each sensor according to measurement time. Appendix A show the mean and standard deviation of the sensor outputs for each sensor, according to the concentration of the substance in the sweat. The slopes of the glucose and lactate sensors measured by the readout PCB using artificial sweat were −1.57 μAmM^−1^ and −5.03 μAdec^−1^, respectively. Amperometric currents of both sensors increased by about 0.3 μAmM^−1^ compared to −1.32 μAmM^−1^ and −4.73 μAdec^−1^ when measured with the commercial potentiostat. The sensitivity of Na^+^ and K^+^ sensors measured by the readout PCB using artificial sweat was measured as 64.75 mVdec^−1^ and 61.82 mVdec^−1^, respectively. When measured with the commercial potentiostat’s potentiometric measurement results of both sensors increased by 0.5 mVdec^−1^, showing a Na^+^ and K^+^ sensitivity of 65.26 mVdec^−1^ and 62.19 mVdec^−1^, respectively. Appendix A shows the resistance change of the temperature sensor according to the measurement time. Appendix A shows the resistance change of the temperature sensor according to the temperature of artificial sweat measured by a commercial temperature sensor. As the temperature of artificial sweat was increased, the resistance of the temperature sensor decreased by an average of 0.25% per 1 °C. This is the same value obtained when using a commercial potentiostat, showing a comparable performance between our developed LIG-based temperature sensor and a commercial temperature sensor.

#### 3.3.2. Real-Time Artificial Sweat Concentration/Temperature Sensing in Calibration Mode

Appendix A show the measured value of each sensor according to time and the concentration of each target or temperature. Table 1 shows the real concentration and temperature of artificial sweat and the measured mean and standard deviation values of the concentration and temperature of each substance measured in artificial sweat with each sensor. The measured values compared to the actual concentration and temperature showed an RSD of less than 3%. All sensors were measured simultaneously during the experiment with artificial sweat.

## 4. Conclusions

In this study, we present a LIG-based multiplexed sensing system capable of simultaneously measuring glucose, lactate, K^+^, and temperature in artificial sweat. This multisensor is disposable and can be easily attached and detached to a multiplexed sensing system board using a connector. In particular, the glucose sensor and lactate sensor showed very high sensitivity and low LOD due to LIG’s high active surface area. This shows that the LIG electrode is a suitable material for electrochemical sensors that require high sensitivity and low LOD compared to the expensive, difficult-to-manufacture, and time-consuming materials used in the past. The multiplexed sensing system showed similar amperometric and potentiometric performance to commercial potentiostats. This research can contribute to the development of next-generation personal medical diagnostic devices through the integration of bio-sensing technology with IoT technology for the ongoing fourth industrial revolution. For future work, it is necessary to develop sensors for additional target analytes that can be measured by amperometry and potentiometry. In addition, multisensing systems with measurement methods other than amperometry and potentiometry must be developed in parallel to the miniaturization of such systems for applications in wearable biosensor technology.

## Figures and Tables

**Figure 1 sensors-24-06945-f001:**
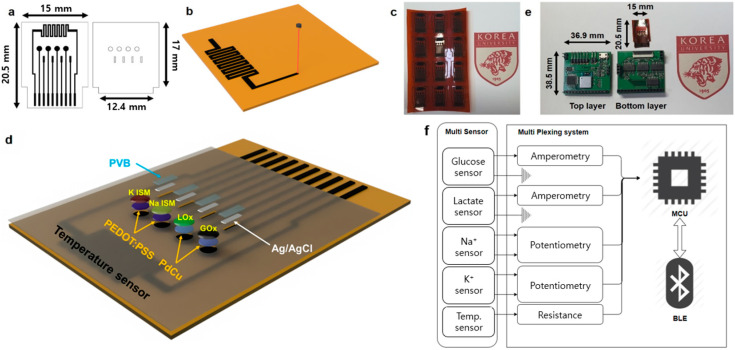
(**a**) *Left:* Multisensor array *Right:* Passivation layer. (**b**) Laser irradiation process of LIG-based electrodes. (**c**) Picture of the multisensor array. (**d**) Structure of multisensor array. (**e**) Picture of multisensor and multiplexed sensing system board. (**f**) Block diagram of multisensor and multiplexed sensing system board.

**Figure 4 sensors-24-06945-f004:**
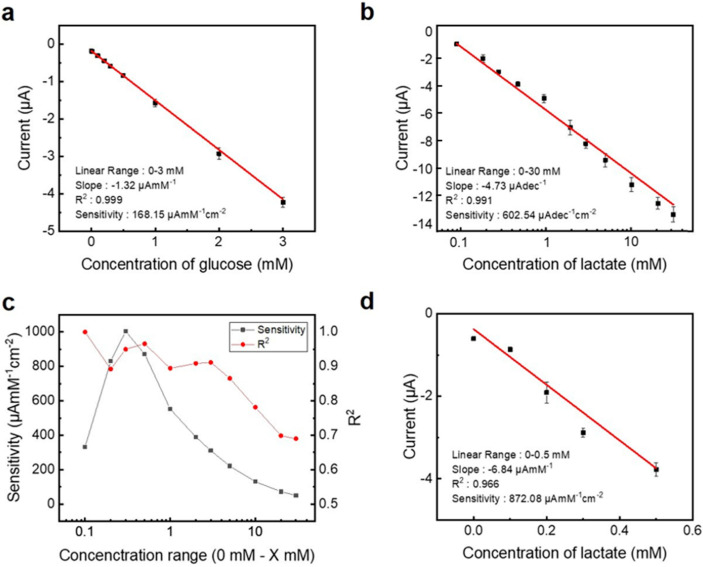
Plot of (**a**) glucose and (**b**) lactate sensor’s amperometric current depending on glucose and lactate concentration. (**c**) Plot of lactate sensor’s sensitivity in the lactate concentration range between 0.1 mM and 10 mM. (**d**) Plot of lactate sensor’s amperometric current in 0–0.5 mM lactate concentration range. Sensitivity and LOD comparison plot of enzymatic electrochemical.

**Figure 5 sensors-24-06945-f005:**
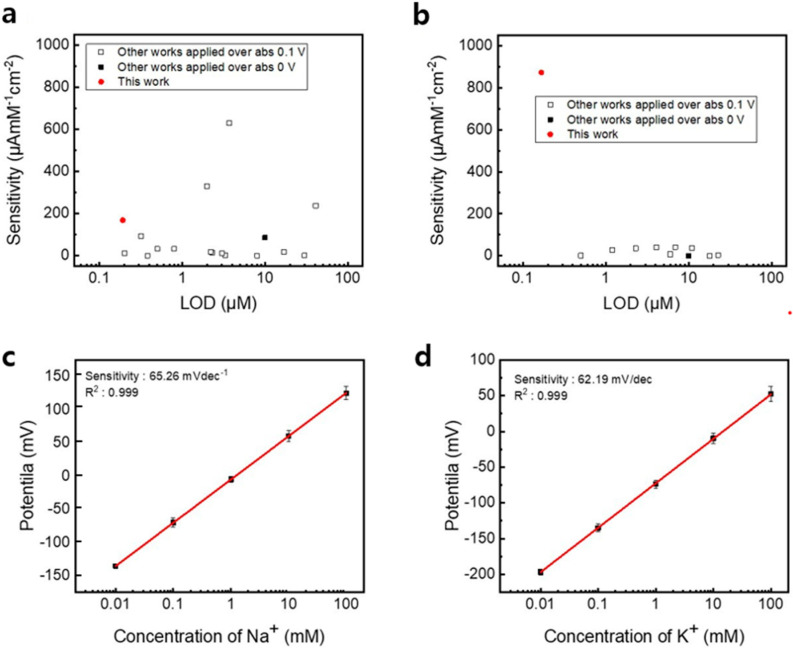
Sensitivity and LOD comparison plot of enzymatic electrochemical (**a**) glucose and (**b**) lactate sensors. Plot of (**c**) Na^+^ and (**d**) K^+^ sensor’s potential according to Na^+^ and K^+^ concentration.

**Figure 6 sensors-24-06945-f006:**
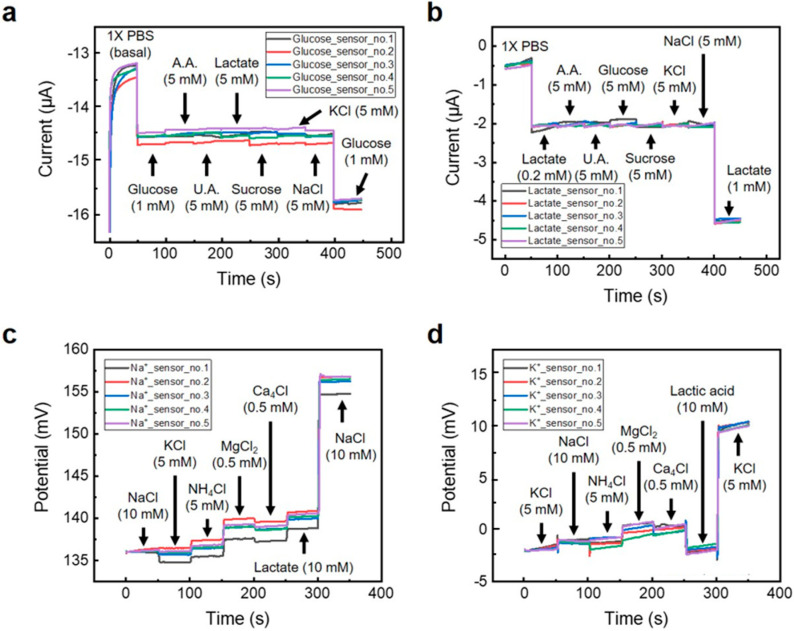
Selectivity plot of (**a**) glucose, (**b**) lactate, (**c**) Na^+^, and (**d**) K^+^ sensor.

**Table 1 sensors-24-06945-t001:** Comparison of measured and actual concentration and temperature measured by multiplexed sensing system in calibration mode.

Sensor	Glucose			Lactate		
Real concentration (μM)	50	100	150	1000	1500	2000
Measurement concentration (μM)	52.23	101.84	146.84	1043.07	1560.25	2049.52
RSD (%)	2.18	0.91	1.06	2.11	1.97	1.22
**Sensor**	**Na^+^**			**K^+^**		
Real concentration (μM)	20	40	80	2	4	8
Measurement concentration (μM)	20.19	39.73	80.51	2.01	3.95	8.06
RSD (%)	0.47	0.34	0.32	0.25	0.63	0.37
**Sensor**	**Temperature**					
Real Temp. (°C)	30.3		35.4		40.2	
Measurement Temp. (°C)	30.43		35.53		40.33	
RSD (%)	0.21		0.18		0.16	

## Data Availability

Data are contained within the article.

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
