# Peer review of "LIG-Based High-Sensitivity Multiplexed Sensing System for Simultaneous Monitoring of Metabolites and Electrolytes"

_sensors, 2024, doi:10.3390/s24216945_

Round 1

Reviewer 1 Report

Comments and Suggestions for Authors

In this work, the authors successfully developed a multifunctional sensor system capable of real-time monitoring of various biochemical markers in artificial sweat. This study includes several intriguing findings, demonstrating that the multi-sensor array prepared using laser-induced graphene (LIG) exhibits excellent selectivity, sensitivity, and accuracy. These findings provide valuable insights for the development of wearable health monitoring devices. I believe this manuscript is suitable for publication in Sensors after minor revisions.

1.          It is recommended to increase the font size in Figure 1f, as the current image is difficult to read.

2.          Figures 3c, 4b, 4c, and 6c lack scale markings on the y-axis. Please review the entire manuscript for consistency.

3.          The description of Figure 3d does not match the pattern in the image. Please ensure that the text corresponds to the figure, and check for any discrepancies in other figures as well.

4.          It is suggested to provide a more detailed explanation of the relationship between current and glucose/lactic acid concentration in Figure 4 and clarify the conclusions drawn from these data.

5.          The temperature module of the multifunctional sensor was tested using only artificial sweat, while the other parameters were tested separately. It is recommended to attempt testing all parameters using a single artificial sweat sample to demonstrate the potential for practical applications.

Author Response

Comments 1 It is recommended to increase the font size in Figure 1f, as the current image is difficult to read.

Response 1: We have increased the font size inside Figure 1f to improve visibility according to the suggestion.

Comments 2. Figures 3c, 4b, 4c, and 6c lack scale markings on the y-axis. Please review the entire manuscript for consistency.

Response 2: We have added scale markings to Figures 3c, 4b, 4c, and 6c, and have also included them in other Figures that were missing scale markings.

Comments 3. The description of Figure 3d does not match the pattern in the image. Please ensure that the text corresponds to the figure, and check for any discrepancies in other figures as well.

Response 3: Since it represents the concentration of NaCl, not Na+, we have revised the x-axis label to 'Concentration of NaCl (mM)’

Comments 4. It is suggested to provide a more detailed explanation of the relationship between current and glucose/lactic acid concentration in Figure 4 and clarify the conclusions drawn from these data.

Response 4: The purpose of Figure 4 is to showcase sensitivity, linear range, limit of detection, and R² of the fabricated sensors. In section 3.2.1 of the first submitted manuscript, the sensitivity selectivity, linear range, and limit of detection of the fabricated LIG-based multi-sensor array have been mentioned, but the R² information was missing in the first submitted manuscript, and hence we have added the R² information to section 3.2.1 of the revised manuscript.

Additionally, Figure 4b and 4d both present the measurement results of the fabricated lactate sensor, but the units for sensitivity differ. Figure 4b shows the measured currents in unit of μAdec-1 while Figure 4d shows the measured currents in unit of μAmM-1. Figure 4b clearly shows how the sensitivity of the fabricated lactate sensor changes as the lactate concentration varies in log scale, showing how the output changes as the input varies by decade in unit of μAdec-1. On the other hand, Figure 4d was made to emphasize the linearity of the fabricated lactate sensor at low lactate concentration ranfw (0 - 0.5 mM), in which the sensitivity is expressed in unit of μAmM-1 and it shows the linear sensitivity with lactate concentration. This information has also been added to the revised manuscript section 3.2.1.

Comments 5. The temperature module of the multifunctional sensor was tested using only artificial sweat, while the other parameters were tested separately. It is recommended to attempt testing all parameters using a single artificial sweat sample to demonstrate the potential for practical applications.

Response 5: We conducted the experiment in three stages. In the first stage, we measured each substance individually using a commercial potentiostat to obtain reference results. In the second stage, we prepared solutions with the target substances and measured them using a multiplexed sensing system. In the third stage, we measured glucose, lactate, Na+, K+, and temperature simultaneously using a multiplexed sensing system with artificial sweat. We believe that the content the reviewer is requesting corresponds to the third stage output, which has already been included in the manuscript in section 3.3.2. Real-time artificial sweat concentration/temperature sensing in calibration mode. There are already quite a few graphs in the manuscript. Therefore, adding more graphs would make the manuscript rather too long and hence some of the measurement result graphs have been placed in the supplementary material. It was mentioned in the manuscript.

To clarify that simultaneous measurements were conducted in artificial sweat, we have added this information to the manuscript.

Reviewer 2 Report

Comments and Suggestions for Authors

In this study, the authors developed a wearable bio-sensing system for non-invasive monitoring of body fluids, integrated with a smartphone app for data processing and wireless communication. The system uses potentiometric and amperometric measurements with calibration for accurate readings. Laser-induced graphene (LIG)-based sensors for glucose, lactate, Na+, K+, and temperature were created, offering a 16-fold increase in active surface area for enhanced performance. The glucose and lactate sensors achieved high sensitivity (168.15 and 872.08 μAmM⁻¹cm⁻²) and low detection limits, while the Na+ and K+ sensors showed sensitivities of 65.26 and 62.19 mV/decade. Temperature sensors accurately measured within 20-40℃.

Although it is an interesting study, but author must address the following major issues to improve the manuscript standards, as suggested below,

        I.            The introduction section does not provide a clear picture of the gap and need of this study even though a lot of sensing devices already have been reported to detect the various types of analytes using graphene and 2D materials as mentioned in these recent reports (doi.org/10.1002/adfm.202204781; https://doi.org/10.1021/acsaelm.3c01045)? A detailed comparison should be added which proves the significance of this study and mentions the above reports in the introduction.

     II.            The quality of the most of the Figures is not good and readable e.g., Fig 1a, Fig 1f . The author should add high-resolution figures.

  III.            The More details about the Raman analysis should be mentioned in the manuscript. At what substrate authors performed the Raman analysis? What is the laser wavelength and power of the incident laser they used in this experiment. How did author claimed that the ratio between 2D/G peak is showing multilayer? It could also be attributed to the defects or non uniform growth.

  IV.            More important, the author should discuss about the glucose and lactate sensors and their selectivity test in detail.  How they changed the concentrations? How stable these testing could be?

Why there is different current response correspond to different substances like NH4Cl, MgCl2, CaCl4? More details should be explained with proper refernces as mentioned in these recent studies 

    V.  Remarks: Major Revision is Required to improve the manuscript quality

Comments on the Quality of English Language

minor language corrections are required.

Author Response

Comments 1. The introduction section does not provide a clear picture of the gap and need of this study even though a lot of sensing devices already have been reported to detect the various types of analytes using graphene and 2D materials as mentioned in these recent reports (doi.org/10.1002/adfm.202204781; https://doi.org/10.10

21/acsaelm.3c01045)? A detailed comparison should be added which proves the significance of this study and mentions the above reports in the introduction.

Response 1: We have mentioned the recommended papers in line 75 of the Introduction and added an explanation to the manuscript, stating that LIG, unlike previous studies, does not require a vacuum process during electrode patterning, allowing for faster electrode fabrication. Additionally, its 3D porous structure increases the active surface area, thereby enhancing sensitivity.

Comments 2. The quality of the most of the Figures is not good and readable e.g., Fig 1a, Fig 1f. The author should add high-resolution figures.

Response 2: We increased the font size in figures including Figure 1a and 1f to improve visibility and also reviewed/adjusted the other figures to improve the readability, enlarging the small tests when they are too small.

Comments 3. The More details about the Raman analysis should be mentioned in the manuscript. At what substrate authors performed the Raman analysis? What is the laser wavelength and power of the incident laser they used in this experiment. How did author claimed that the ratio between 2D/G peak is showing multilayer? It could also be m attributed to the defects or non uniform growth.

Response 3: It is well known that, in Raman spectroscopy, the number of graphene layers is typically calculated using the I2D/IG ratio[1]. As the number of graphene layers increases, the intensity of IG increases, reducing the I2D/IG ratio, which is similar to the shape of the graph in Figure 3a. Therefore, we believe that the graphene is composed of a few layers. The peak related to defects mentioned by the reviewer is the D peak, and as the number of defects decreases, the D peak also reduces. In Figure 3a, the D peak is visible, indicating the presence of defects. However, we believe this occurred because we used raster mode to increase the active surface area, and the peak values likely resulted from localized overheating and damage to the electrode surface during laser processing.

Additionally, the laser irradiation conditions are described in line 107 of Section 2.1.1. Design and fabrication of multi-sensor array and passivation layer, as follows. The wavelength of 10.6 µm was used for laser irradiation in the experiment and it was described in the revised manuscript.

(LIG electrodes were formed by irradiating a 250 μm-thick PI film (DuPont inc., USA) with a CO2 laser (VLS2.30, Universal Laser System Inc., USA) using raster mode at 1,000 PPI (pulse per inch), 1000 DPI(dot per inch), a power of 4.2 W, wavelength of 10.6 µm, and scan speed of 88.9 mm/s.)

We also have added, in section 3.2.1 of the revised manuscript, reference [54] where the meaning and the usage of the I2D/IG ratio have been mentioned previously.

Comments 4. More important, the author should discuss about the glucose and lactate sensors and their selectivity test in detail. How they changed the concentrations? How stable these testing could be?

Why there is different current response correspond to different substances like NH4Cl, MgCl2, CaCl4? More details should be explained with proper refernces as mentioned in these recent studies

Response 4: During the selectivity experiments, we aimed to maintain analytes’ concentrations similar to those found in human sweat to verify whether the fabricated sensor shows selectivity under those conditions[2,3]. We maintained the concentrations of NH4Cl, MgCl2, CaCl4 similar to those found in the references, even if the concentrations of interfering substances may be slightly higher and the target substances are slightly lower. The reason for this difference is that we determined the sensor should not react to irrelevant substances that should not be measured, even if their concentrations are higher, while it should still detect the target substances even at lower concentrations.

We have added an explanation regarding the different concentrations of interfering substances, along with the reference literature, to section 3.2.2 of the revised manuscript.

Comments 5. Remarks: Major Revision is Required to improve the manuscript quality

Response 5: Thank you for the prompt review and useful comments for revising the manuscript to improve the manuscript. We have addressed and completed all the modifications as much as we can, based on the reviewers’ comments.

Reference

  1. Shanmugam, M.; Jacobs-Gedrim, R.; Song, E.S.; Yu, B. Two-Dimensional Layered Semiconductor/Graphene Heterostructures for Solar Photovoltaic Applications. Nanoscale 2014, 6, 12682–12689, doi:10.1039/c4nr03334e.
  2. Gualandi, I.; Tessarolo, M.; Mariani, F.; Possanzini, L.; Scavetta, E.; Fraboni, B. Textile Chemical Sensors Based on Conductive Polymers for the Analysis of Sweat. Polymers (Basel). 2021, 13, doi:10.3390/polym13060894.
  3. Maughan, R.J.; Watson, P.; Shirreffs, S.M. Implications of Active Lifestyles and Environmental Factors for Water Needs and Consequences of Failure to Meet Those Needs. Nutr. Rev. 2015, 73, 130–140, doi:10.1093/nutrit/nuv051.

Round 2

Reviewer 2 Report

Comments and Suggestions for Authors

Accept it in the present form.